# Correlations of Salivary and Blood Glucose Levels among Six Saliva Collection Methods

**DOI:** 10.3390/ijerph19074122

**Published:** 2022-03-30

**Authors:** Yangyang Cui, Hankun Zhang, Jia Zhu, Zhenhua Liao, Song Wang, Weiqiang Liu

**Affiliations:** 1Tsinghua Shenzhen International Graduate School, Tsinghua University, Shenzhen 518055, China; cuiyy20@mails.tsinghua.edu.cn (Y.C.); zhanghk20@mails.tsinghua.edu.cn (H.Z.); zhuj@tsinghua-sz.org (J.Z.); 2Department of Mechanical Engineering, Tsinghua University, Beijing 100084, China; 3Biomechanics and Biotechnology Lab, Research Institute of Tsinghua University in Shenzhen, Shenzhen 518057, China; liaozh@tsinghua-sz.org

**Keywords:** saliva collection method, salivary glucose, blood glucose, diabetes mellitus, correlation

## Abstract

Background: Saliva has been studied as a better indicator of disorders and diseases than blood. Specifically, the salivary glucose level is considered to be an indicator of diabetes mellitus (DM). However, saliva collection methods can affect the salivary glucose level, thereby affecting the correlation between salivary glucose and blood glucose. Therefore, this study aims to identify an ideal saliva collection method and to use this method to determine the population and individual correlations between salivary glucose and blood glucose levels in DM patients and healthy controls. Finally, an analysis of the stability of the individual correlations is conducted. Methods: This study included 40 age-matched DM patients and 40 healthy controls. In the fasting state, saliva was collected using six saliva collection methods, venous blood was collected simultaneously from each study participant, and both samples were analyzed at the same time using glucose oxidase peroxidase. A total of 20 DM patients and 20 healthy controls were arbitrarily selected from the above participants for one week of daily testing. The correlations between salivary glucose and blood glucose before and after breakfast were analyzed. Finally, 10 DM patients and 10 healthy controls were arbitrarily selected for one month of daily testing to analyze the stability of individual correlations. Results: Salivary glucose levels were higher in DM patients than healthy controls for the six saliva collection methods. Compared with unstimulated saliva, stimulated saliva had decreased glucose level and increased salivary flow. In addition, unstimulated parotid salivary glucose was most correlated with blood glucose level (R^2^ = 0.9153), and the ROC curve area was 0.9316, which could accurately distinguish DM patients. Finally, it was found that the correlations between salivary glucose and blood glucose in different DM patients were quite different. The average correlation before breakfast was 0.83, and the average correlation after breakfast was 0.77. The coefficient of variation of the correlation coefficient before breakfast within 1 month was less than 5%. Conclusion: Unstimulated parotid salivary glucose level is the highest and is most correlated with blood glucose level, which can be accurately used to distinguish DM patients. Meanwhile, the correlation between salivary glucose and blood glucose was found to be relatively high and stable before breakfast. In general, the unstimulated parotid salivary glucose before breakfast presents an ideal saliva collecting method with which to replace blood-glucose use to detect DM, which provides a reference for the prediction of DM.

## 1. Introduction

With the advancement of living standards and environmental changes, the incidence of diabetes mellitus (DM) is increasing daily, which has steadily developed into one of the major threats to the health and well-being of modern people, imposing a significant financial burden on society [1]. According to data released by the International Diabetes Federation (IDF) in 2019, about 4.2 million adults died from DM and its complications, which is equivalent to 1 death every 8 s [2]. It is expected that by the year 2045, a staggering 700 million adults across the world will be living with some types of DM [3]. Today, around half of the 463 million adults with DM are unaware of their illness, placing them at an increased risk of acquiring catastrophic DM-related complications. [4]. Additionally, more recent research indicates that DM may be present for up to seven years prior to clinical diagnosis [4,5]. In this timeframe, people may develop life-threatening problems over time, including retinal vascular disease, foot ulcers, renal failure, and other types of multiple organ damage [6]. However, the origin and pathogenesis of DM remain unknown, and there are no therapies available as of yet. The primary clinical control strategies include the self-monitoring of blood glucose, medical nutrition therapy, exercise therapy, DM patient education, and DM patient medication therapy. Self-monitoring of blood glucose is critical for DM control and serves as the foundation for all strategies [7]. Sim et al. also [8] point out that the frequent monitoring of blood glucose is of great significance for patients in order to manage the condition and control the development of their complications. Unfortunately, traditional blood glucose monitoring methods usually require blood sampling operations. Blood extraction is painful, inconvenient and presents the risk of infection and mental pains to patients, especially for those patients who need to check their blood glucose levels several times a day [9].

Not only does an urgent need exist for a non-invasive glucose monitoring technology so as to make a major improvement in the lives of millions of people around the world living with DM, but also to ease preventive monitoring [10]. Over the past few decades, the non-invasive monitoring of DM by fluid analysis (as well as other biological fluids such as urine, sweat, and saliva) has attracted worldwide attention [11,12,13]. Among the existing non-invasive methods, salivary glucose has a very positive significance, and has received much attention [14,15,16]. Saliva is considered an ultrafiltrate of blood and is a potential source of clinical information that accurately reflects the pathological state, according to the literature. The literature suggests that saliva is considered an ultrafiltrate of blood and is a potential source of clinical information that can fully reflect the pathological state [17,18]. Furthermore, saliva has numerous advantages as a diagnostic fluid because it is simple to collect, preserve, and contains exceedingly high-quality DNA. Thus, saliva may be an excellent substitute for blood. [19]. Caixeta et al. [20] demonstrated that salivary glucose was a potential outcome for screening, diagnosing, and monitoring DM, as they found a 95.2% accuracy of the laboratory test of salivary glucose in response to blood glucose. According to Rodrigues et al. [21], saliva contains biomarkers such as different proteins, fatty acids, and carbs that, similarly to blood, can reflect changes in human physiological activities, and so may serve as an alternative source for the early identification and monitoring of DM.

Although the biomarkers in saliva reflect the health status of the human body, the use of salivary glucose as a diagnostic fluid for DM has been hindered, mainly because the correlation between salivary glucose and blood glucose has been greatly controversial [22]. Studies have shown that for DM patients, the salivary glucose level is positively correlated with the blood glucose level, so salivary glucose can be used as a marker for DM detection [23,24]. However, in different studies, the relationship between salivary glucose and blood glucose is quite different with completely opposite conclusions, pointing out that salivary glucose and blood glucose have no significant correlation [25,26]. The key factor contributing to this phenomenon is that saliva is collected differently, and most studies view saliva wrongly as a homogeneous body fluid. However, saliva cannot be considered a solitary fluid, as it consists of a complex mixture that comprises the secretions of three major glands (the parotid, submandibular and sublingual glands, each secreting a characteristic type of saliva), hundreds of minor salivary glands, gingival crevicular fluid and debris [27]. Moreover, saliva composition is constantly changing and the composition is affected, among other things, using saliva collection methods and general health. The saliva was mainly divided into whole saliva and single gland saliva, and saliva collection includes both stimulated and unstimulated methods. Considering all this sampling-related composition variability, we can confirm that the saliva collection method and location will certainly have a great influence on the salivary glucose level [28,29].

Therefore, in this study, whole saliva, parotid saliva and sublingual/mandibular saliva were collected with both stimulated and unstimulated methods, from 40 age-matched DM patients and 40 healthy controls under a fasting state in the morning. Saliva and venous blood were collected simultaneously from each study participant, and both samples were analyzed at the same time using the glucose oxidase peroxidase (GOD-POD) method. Saliva flow rate, salivary glucose level and the correlation between salivary glucose and blood glucose were calculated. The ideal saliva collection method was determined, which laid the foundation for the detection of salivary glucose. Meanwhile, the receiver operating characteristic curve (ROC) method was used to analyze the diagnostic value of saliva glucose detection in DM, and its application in the diagnosis of DM was discussed. Finally, the individual correlation and the stability of the individual correlation were analyzed.

## 2. Materials and Methods

### 2.1. Ethics Statement

Human saliva and blood were collected from donors who provided written informed consent. The collection of human saliva samples was approved by the local ethics committee at Tsinghua University.

### 2.2. Participants

In this study, a total of 80 participants were included, 40 of whom were DM patients and 40 were healthy controls. The inclusion criteria were as follows: participants must be in good general health and ≥18 years old. On the day of collection, all participants were fever-free and maintained excellent dental hygiene. If oral inspection revealed poor oral hygiene, hyposalivation, oral complaints, or other oral disorders (e.g., mucosal lesions, clinical evidence of ongoing periodontal disease), participants were immediately removed from continued participation in the study.

Firstly, saliva and blood were collected from 80 participants, and the population correlation analysis was carried out to determine the ideal saliva collection method in the morning (7:30–8:00). Secondly, 20 DM patients and 20 healthy controls were arbitrarily selected from the 40 individuals for each group to collect saliva and blood every day (before and after breakfast) for one week with the determined saliva collection method, and to conduct an individual correlation study. Finally, 10 DM patients and 10 controls were arbitrarily selected from the remaining 20 individuals for each group for continuous monitoring every day (before breakfast) for one month, the weekly correlation of the individuals was calculated, and the coefficient of variation (CV) was calculated to analyze the stability of the individual correlation. An initial overall screening of the sampling possibilities is to choose the ideal saliva collection method, with a further study using the most promising method to determine the correlation of individual relationships and their stability. The correlation in this study refers to the correlation between blood and saliva glucose.

### 2.3. Samples Collection

The participants were asked to avoid smoking, brushing teeth and to not eat or drink during the 30 min prior to sample collection. Before sampling, the mouth was rinsed with water before collection to remove food residues in the oral cavity [30]. A salivette (Sarstedt, 51.5134) (including untreated swabs and swabs stimulated by citric acid) was used to collect saliva, with six collection methods (as shown in Figure 1).

For each participant, whole saliva, parotid saliva and sublingual/mandibular saliva were collected with both stimulated and unstimulated methods. Each type of sample is respectively denoted as unstimulated whole saliva (UWS), stimulated whole saliva (SWS), unstimulated sublingual/submandibular saliva (USS), stimulated sublingual/submandibular saliva (SSS), stimulated parotid saliva (SPS) and unstimulated parotid saliva (UPS). The collection time was 7:30–8:30 in the morning. All saliva samples were obtained sequentially in the same clinical room. All samples were collected in pre-chilled polypropylene tubes on ice to prevent the degradation of sensitive peptides. Saliva was obtained using all three procedures and totaled 5 mL. Finally, it was brought to the laboratory on a routine basis and centrifuged at −20 °C for future use.

Venous blood was drawn from all subjects following the last saliva sample. The samples were gently mixed for 1 min and then placed immediately on ice for 30 min. After centrifuging the samples at 1000× *g* for 15 min at 4 °C, the upper two-thirds aliquot of plasma was frozen at −70 °C until analysis. In the individual correlation study, UPS and blood were collected before and after breakfast, other processing conditions and storage methods were unchanged, and they were collected daily for one week. Conversely, for the individual correlation stability study, UPS and blood were collected before breakfast, while other processing conditions and storage methods remained unchanged, and were collected daily for one month. Glucose and saliva glucose were measured by GOD-POD (Glucose kit, Beijing Furui Runkang Biotechnology Co., Ltd., Beijing, China).

### 2.4. Statistics

To perform the statistical analysis, Graphpad 8.0 (GraphPad Software, San Diego, CA, USA) was employed. The data from the enumeration were expressed as relative numbers, and the $\chi^{2}$ test was used to compare groups. The measurement data were normally distributed and reported as (mean standard deviation $\bar{X}\pm SD$), with the *t* test so as to compare groups. The salivary glucose levels of different genders and ages were determined in each group using the *t*-test for two independent samples. The receiver operating characteristic (ROC) approach was utilized in this study to completely evaluate the diagnostic utility of salivary glucose detection in DM. Furthermore, *p* < 0.05 indicates a statistically significant difference.

## 3. Results

### 3.1. Sample Characteristics

There were 14 males and 26 females with an average age of (50.1 ± 4.8) years in the patient group and 14 males and 26 females with an average age of (49.7 ± 3.7) years in the control group. No statistically significant differences were observed due to gender (t = 0.641, *p* = 0.289) or age (t = 0.181, *p* = 0.510) between the two groups.

The blood and saliva flow rates of the studied groups are shown in Table 1. As can be observed, the DM patient has a much lower saliva flow rate than the control group, and stimulated saliva has a significantly greater saliva flow rate than unstimulated saliva. Additionally, the patient group had greater salivary glucose levels (average levels) than the control group in all six saliva types (as shown in Table 2), with the highest level of UPS glucose and the lowest level of stimulated entire salivary glucose, as illustrated in Figure 2. The patient group’s mean blood glucose and salivary glucose levels were higher than the control group. Salivary glucose levels in the patient group ranged from 0.67 to 4.31 mmol/L. Salivary glucose levels in the control group ranged between 0.51 and 3.02 mmol/L. 

### 3.2. Validation of Diagnostic Performance by ROC

Figure 3 showed the ROC curve of salivary glucose in the diagnosis of DM. It can be seen that the salivary glucose obtained using the unstimulated method can significantly distinguish the DM patients from the control group, and the *p* values were all < 0.001. Among them, UPS can be better used to diagnose DM, and its area under the curve (AUC) was 0.9316 (excellent accuracy). The best cut-off value to distinguish between DM and controls was 2.045 mmol/L, with sensitivity at 82.6% and specificity at 86.6%. The second was USS (AUC = 0.8988), and the predictive ability of UWS and USS was basically the same (AUC = 0.8969). The ROC values of saliva collected using the stimulated methods were all lower. In conclusion, the salivary glucose level obtained by UPS was the highest, and can be better used for the diagnosis of DM. Therefore, UPS saliva was used to study the correlation between blood glucose and salivary glucose.

### 3.3. Correlation Studies

#### 3.3.1. Population Correlations

In both DM patients and controls, there was a substantial association between blood glucose and unstimulated salivary glucose levels (*p* < 0.001). Among them, the UPS glucose level was found to be the most correlated with the blood glucose level, R^2^ = 0.9153, and the regression equation was Y = 0.3626 ∗ X − 0.3282 (as shown in Figure 4). Salivary glucose levels increased in both the patient and control groups in correlation with blood glucose levels, as shown in Figure 4.

#### 3.3.2. Individual Correlations

It can be seen from the population correlation that the UPS had the highest glucose level and was most correlated with the blood glucose level. Besides, it can be better used to diagnose DM. Therefore, the UPS was used to study the individual correlation between blood glucose and salivary glucose. The average correlation coefficient between pre-prandial salivary glucose and blood glucose in DM patients was 0.88, and the average correlation coefficient between postprandial salivary glucose and blood glucose was 0.813, while the average correlation coefficient between pre-prandial salivary glucose and blood glucose in the control group was 0.78. The average correlation coefficient between postprandial salivary glucose and blood glucose was 0.7325. As shown in Figure 5, while the correlation coefficients before and after breakfast for various individuals within a week varied significantly, the overall consistency was high. In general, the correlation coefficients before breakfast were higher than those after breakfast.

#### 3.3.3. Stability of Individual Correlations

As illustrated in Section 3.3.2, we can conclude that the correlation before breakfast was stronger and more stable in the individual correlation study. Therefore, the saliva before breakfast was used to examine the stability of the correlation coefficient. The average daily pre-breakfast salivary glucose and blood glucose levels were tracked consistently for one month under the assumption that all other experimental circumstances and techniques were constant. Finally, for each subject, the correlation coefficient between pre-prandial salivary glucose and blood glucose levels within a week was computed. Table 3 summarizes the specific findings. As can be observed, all results had a CV < 5%.

## 4. Discussion

Saliva is considered as an ultrafiltrate of blood and can replace blood for the monitoring of DM [17]. Among all salivary parameters, salivary glucose is most closely associated with the oral environment of DM patients [31]. However, the existence and degree of correlation between salivary glucose and blood glucose has been controversial. The main reason for this controversy is the influence of salivary glucose levels. The following factors are of interest in this regard:

First, the saliva collection method will affect the salivary glucose level and the saliva flow rate, which was confirmed in our previous study. This study pointed out that the saliva collection method is an important factor affecting saliva glucose levels and the saliva flow rate. Moreover, we found that UPS was the most correlated with blood glucose, which provided a reference for the prediction of DM [17]. However, in our previous study, only for healthy people were included, while the saliva composition of DM patients is more complex, making it impossible to confirm whether the conclusions reached in that study are also applicable to DM patients.

Second, the population correlation can only reflect the general situation of the relationship between group blood glucose and salivary glucose. If it is used for the monitoring of the correlation between individual salivary glucose and blood glucose, its reliability is not verified, and it becomes difficult to accurately reflect the blood glucose levels of different individuals at different stages. This is another factor that was not thoroughly investigated in our previous study [17]. Therefore, in order to solve the above problems, this study compared the saliva flow rate and saliva glucose level of six different saliva collection methods in DM patients for the first time and explored the population correlation of saliva glucose and blood glucose. Furthermore, the saliva before and after breakfast were collected every day for one week to obtain the individual correlation. Finally, a month of follow-up monitoring was carried out to determine whether the individual correlation was stable, which laid a more scientific foundation for better monitoring the blood glucose level value with the salivary glucose level.

In this study, salivary glucose was found in both DM patients and healthy controls, which is consistent with the observations of Ephraim et al. [32] who also found salivary glucose for both groups. However, Amer et al. [33] found no salivary glucose of the control group. We found that the saliva glucose levels of DM patients obtained by the six saliva collection methods were higher than in the control group, and the difference was statistically significant, which was consistent with the conclusions of most other studies [23,24]. Similar to our study, Mishra et al. [34] found a positive and statistically significant correlation between salivary and blood glucose in DM patients. Therefore, salivary glucose can be used to predict blood glucose level in DM patients. Karjalainen et al. [35] demonstrated that after good blood glucose control in DM patients, both salivary glucose and blood glucose are reduced to varying degrees, which suggests that salivary glucose and blood glucose have a certain correlation.

Zolotukhin et al. [36] revealed that salivary glands operate as blood glucose filters, and that hormones and neuromodulation can affect blood glucose levels. Abikshyeet et al. [37] showed that glucose leakage from salivary gland duct cells increased in DM patients, and therefore salivary glucose levels increased in DM patients. This is attributed to microvascular abnormalities in the blood vessels and basement membrane modifications of DM patients. Hyperglycemia promotes the production of advanced glycation end products (AGEs). The AGEs crosslink collagen and extracellular matrix proteins, among other things, causing alterations in the basement membrane and endothelial dysfunction, thereby increasing their permeability, which accounts for the increased glucose entry from the blood into saliva in DM patients [38]. This is supported by Belazi et al. [39] who proposed that the increased permeability of the basement membrane in DM patients may result in the leakage of smaller molecules such as glucose into the whole saliva via the gingival gap. Consequently, elevated glucose levels in the salivary secretions of DM patients have been reported. The presence and increase in glucose levels in saliva are multifactorial, and no single mechanism exists in DM patients and controls. Therefore, the association between salivary glucose and blood glucose established in this study can be deemed trustworthy.

However, discrepancies in the outcomes of different studies may reflect varied study designs, as well as a variety of methodologies and sample selection criteria [40]. A number of factors influence the salivary glucose level. The saliva collection method is the most essential [17]. Saliva samples taken in both stimulated and unstimulated situations included total saliva, parotid saliva, and mandibular/sublingual gland saliva. This study used six different methods to collect saliva and discovered that unstimulated saliva had higher glucose levels than stimulated saliva. Second, in terms of blood glucose levels, parotid salivary glucose is superior to entire saliva and to the sublingual gland. Takeda et al. [41] examined the chemical levels of saliva in healthy persons under various situations and discovered that practically all metabolites were greater in unstimulated saliva than stimulated saliva. Jha et al. [42] also noticed this. They discovered that mean salivary glucose levels were greater in unstimulated saliva from control and DM patients than in stimulated saliva, which is consistent with the current study’s findings. This is related to the aqueous phase, which increases salivary secretion, resulting in a dilution of the molecule of interest’s level [43].

In this study, we discovered that glucose levels in the parotid salivary gland were greater in both the DM patients and controls than in the whole salivary and mandibular/sublingual glands. DM patients’ blood glucose and parotid salivary glucose levels were substantially higher than those of the control group, and the parotid salivary glucose level was strongly associated with the DM patients’ blood glucose level. This is consistent with previous research [44,45]. Our study showed a significantly strong correlation between UPS and blood glucose levels, whereas there was a significant correlation between whole saliva and blood glucose levels. UPS can provide a more accurate representation of blood glucose levels. It could be attributed to an increased permeability of the parotid basal cell membrane in DM patients, resulting in significant glucose release in the parotid duct [46]. This could explain the higher glucose levels in DM patients’ parotid saliva. As a result, glucose levels in parotid saliva may better mirror blood glucose levels than glucose levels in entire saliva. This could be due to the fact that many unknown variables and unstable substances can influence the properties of entire saliva. Saliva taken directly from a single gland is stable and unaffected by oral cavity conditions, thus providing an accurate reflection of the blood glucose status.

The proportionate contribution of various glands to total saliva changes depends on the collection method, stimulation level, age, and even the time of day [47]. To minimize the impact of these characteristics on flow rates and salivary glucose levels, which exhibited considerable disparities in levels as previously reported and directly connected to salivary collection methods [48], participants in this study were sex-matched and closely related to age. Because salivary secretion is so unpredictable, researchers may need to take a different approach when looking into its ingredients or their potential significance as indicators of various physiological states. Although there is now a large body of knowledge about the diagnostic potential of saliva, there are currently no standardized protocols for collecting saliva samples. Different sampling methods are frequently utilized in different studies, and many articles mention little or no patient preparation or sampling procedures [17]. Furthermore, participant characterisation is frequently insufficient without a thorough clinical assessment. The majority of saliva glucose research focuses on whole saliva [18] because it is simple to obtain by spitting into a tube or allowing it to flow from one’s mouth. Saliva obtained from various salivary glands has received little attention. This is the first study to compare whole salivary and glandular salivary glucose, to the best of our knowledge. The findings revealed that the various collection methods resulted in significant changes in salivary glucose levels. The correlation between samples collected using the same procedure and those collected using other methods was closer. Because the parotid gland’s primary role is to digest carbohydrates and create food boluses, serous cells predominate, allowing the gland to generate primarily serous secretions that are high in salivary alpha-amylase and electrolytes. This could account for the increased glucose levels in parotid saliva [49]. Sublingual saliva’s primary role is to lubricate the mouth and protect it from chemical and mechanical impacts. Saliva is mostly released by mucous cells; therefore, it is heavy in mucins and other glycoproteins, which could explain why sublingual glucose levels are higher than in entire saliva [50].

On the surface, saliva collection appears to be a simple, non-invasive technique; however, different collection methods have limits that are not always obvious. A range of organic and inorganic compounds are known to be secreted by the parotid gland. The saliva component is extremely viscous and simple to digest. However, UPS flow rates are relatively modest, and its collection takes some time. Because the submandibular and sublingual glands are so close to one another, they are commonly lumped together. As a result, it is difficult to reliably isolate saliva from these glands. As a result, saliva was collected from both glands at the same time in the current investigation. We further investigated its diagnostic and predictive ability based on the robust and significant association discovered between salivary glucose and blood glucose levels.

We observed that salivary glucose could significantly discriminate between the control and DM patients, where salivary glucose levels obtained from UPS could be better used to diagnose DM with an excellent accuracy, followed by USS, the prediction ability of UWS was basically the same in USS, but the ROC values of saliva collected by stimulated method were all lower. This study shows an AUC of the UPS close to 1 with good specificity and sensitivity, suggesting that UPS glucose is a reliable diagnostic test. The performed ROC analysis allowed for the validation of the validity of saliva in the diagnosis of DM, in agreement with existing studies [51]. In this study, the sensitivity and specificity were 82.6% and 86.6%, respectively, and the overall accuracy was 93.16% for blood glucose values ≥ 2.045 mmol/h. In a study of DM patients in an Indian population by Shashikanth et al. [52], the use of salivary glucose yielded 100% sensitivity, 78% specificity, and 89% AUC. Compared with our study, the AUC reported by Shashikanth is similar. However, there is disagreement regarding the percentage sensitivity and specificity in this study compared to that reported by Shashikanth et al. [52]. We speculate that this may be due to differences in factors such as fasting time, sample size, and different sample estimates.

This study also identified a decrease in unstimulated total salivary flow rate in DM patients, which is consistent with previous studies [53,54]. DM is known to affect the sympathetic and parasympathetic nervous systems of the salivary glands, resulting in decreased salivation, microangiopathy, dehydration, and hormonal changes, which may lead to decreased salivary flow [53]. In previous studies on salivary glucose and blood glucose levels, researchers have studied the population correlation between salivary glucose and blood glucose, and the results cannot represent the individual level. It should be noted that the content of saliva glucose is low, and its level is also affected by many factors. There are individual differences between people, so the difference in salivary glucose level between individuals is also large. Therefore, this study explored the correlation coefficient between the salivary glucose and blood glucose of a single individual.

In order to evaluate the relationship between saliva glucose level and blood glucose level among individuals, this study used 1 week as a time period, adopted the method of continuous detection of pre-prandial and postprandial blood glucose and saliva glucose, and counted the daily statistics for each individual. The correlation coefficient of pre-meal and post-meal was also calculated. This study found that the individual correlation coefficients were high and low. The high correlation value was 0.97, and the low one was only 0.66. It may be due to the different course of DM, the use of insulin and hypoglycemic drugs during monitoring, and the concomitant retinopathy and neurology [54]. In the process of monitoring, other drugs such as softening blood vessels are still used continuously, so the monitoring results are not ideal, but from another perspective, it is also confirmed that the salivary glucose level of different individuals is related to blood glucose level. There are differences in sex, so it is necessary and scientific to establish an individual correlation coefficient. In order to further study the stability of the correlation between the individual salivary glucose level and blood glucose level, this study randomly selected 20 participants from a one-month continuous follow-up monitoring. From the results of this study, using data comparison and analysis, it can be proved that although the correlation coefficient between the individual salivary glucose level and blood glucose level is different and changed slightly, and that the preprandial salivary glucose level is closely related to the blood glucose level. The results have a CV < 5%, which indicates that the overall trend is relatively stable.

Our study has a number of limitations, one of which is the study’s small sample size. Additional studies with larger sample sizes are necessary to confirm the link between blood glucose and salivary glucose in order to develop a saliva-based diagnostic for the diagnosis of DM. Additionally, there are other concerns that need to be answered and studied further in this study. For instance, how to increase the sensitivity of glucose detection, given that salivary glucose levels are lower than blood glucose levels. As a result, it is critical to develop more sensitive technologies to detect salivary glucose that are suited for self-management of DM patients. In conclusion, this study may provide new perspectives for further investigation of the diagnostic potential of salivary glucose as a non-invasive means of monitoring blood glucose. The findings of this study demonstrated that varied methods of saliva collection resulted in significant changes in salivary glucose levels. A comparison of unstimulated and stimulated saliva-collection methods demonstrates that UPS may be the preferable collection method due to its simplicity and relevance. Thus, salivary glucose levels varied significantly between techniques. The findings emphasize the critical nature of consistency when collecting saliva samples, which should exceed the collection method itself. Meanwhile, the correlation coefficient between pre-meal salivary glucose and blood glucose levels is quite high and consistent in DM patients.

## 5. Conclusions

In this study, the salivary flow rate and salivary glucose levels of six different salivary collection methods were compared, and the correlations between individual salivary glucose and blood glucose were explored. A month-long follow-up monitoring yielded the stability of individual correlations. For all six saliva collection methods, the mean salivary glucose levels in DM patients were greater than those in the control group. When comparing stimulated and unstimulated saliva, stimulated saliva glucose levels decreased and the saliva flow increased. It was found that the UPS glucose level was most correlated with blood glucose levels. The AUC was 0.9316, which could accurately distinguish DM patients. The correlation coefficient between saliva glucose and blood glucose in different DM patients was quite different. The average pre-prandial correlation coefficient was 0.83, and the postprandial correlation coefficient was 0.77. Besides, the pre-prandial correlation coefficient had a CV of < 5% within 1 month. In conclusion, based on the findings of this study, it can be inferred that UPS before breakfast may serve as a potential non-invasive adjunct to monitoring blood glucose in DM patients.

## Figures and Tables

**Figure 1 ijerph-19-04122-f001:**
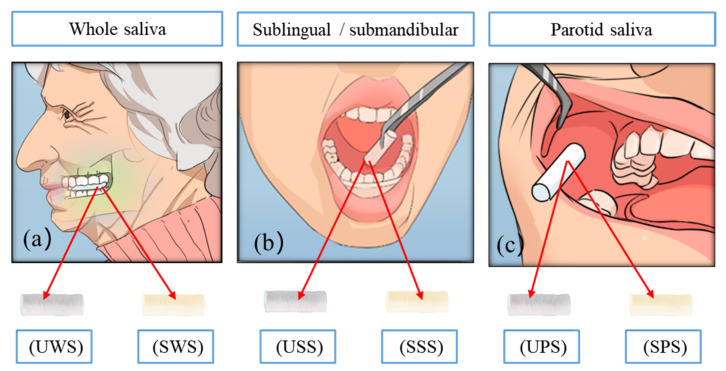
Six methods for collecting saliva samples, (**a**) UWS/SWS: the swab in the test tube was taken out and put in the mouth to chew for 3 min, (**b**) USS/SSS: the swab was put under the tongue and it was taken out after 3 min, (**c**) UPS/SPS: the swab was placed near the left parotid duct and it was taken out after 3 min.

**Figure 2 ijerph-19-04122-f002:**
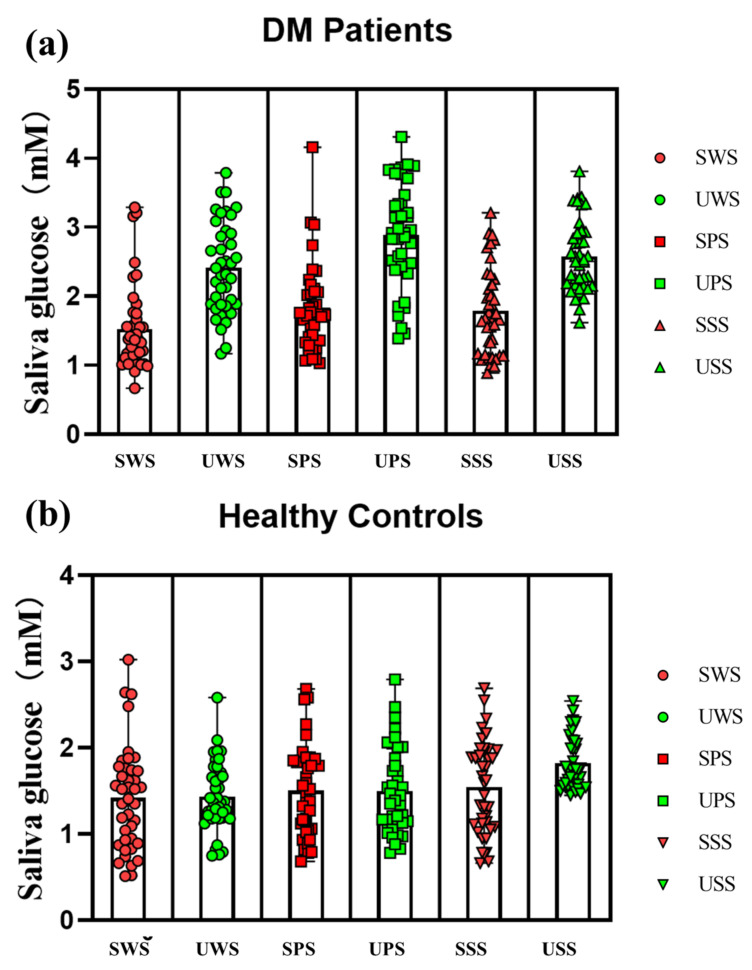
Salivary glucose levels in all participants, (**a**) DM patient group, (**b**) Healthy control group.

**Figure 3 ijerph-19-04122-f003:**
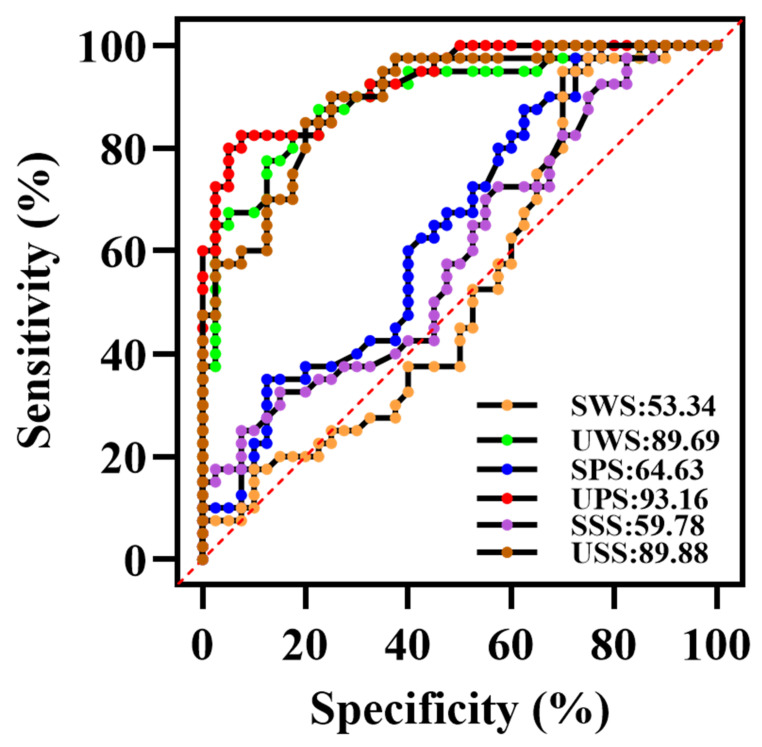
The ROC curve of salivary glucose in the diagnosis of DM of six saliva collection methods.

**Figure 4 ijerph-19-04122-f004:**
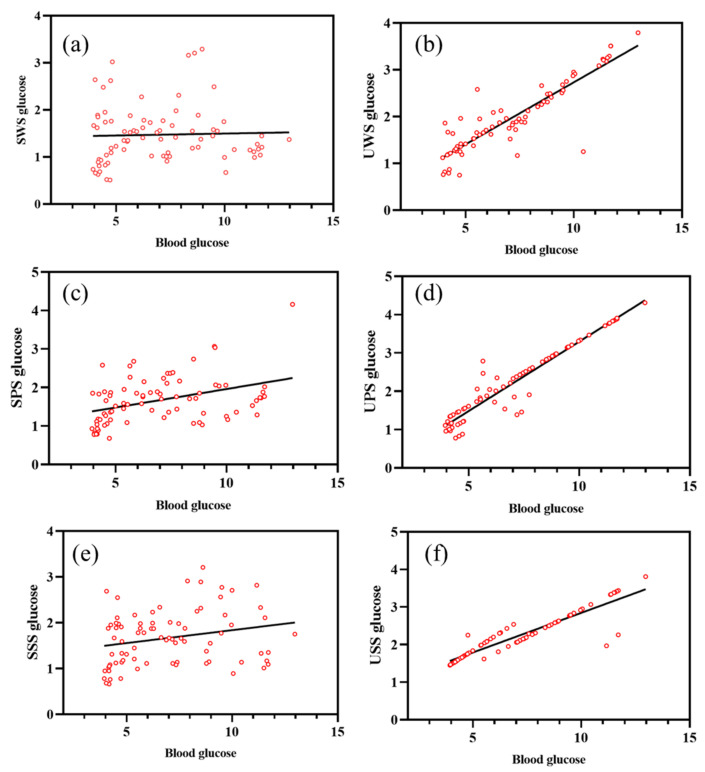
Correlation between salivary glucose of six saliva collection method and blood glucose. (**a**) The correlation between SWS glucose with blood glucose (R^2^ = 0.001127, Sy. x = 0.6183), (**b**) The correlation between UWS glucose with blood glucose (R^2^ = 0.8109, Sy. x = 0.3217), (**c**) The correlation between SPS glucose with blood glucose (R^2^ = 0.1568, Sy. x = 0.5597), (**d**) The correlation between UPS glucose with blood glucose (R^2^ = 0.9153 (maximum correlation), Sy. x = 0.2772), (**e**) The correlation between SSS glucose with blood glucose (R^2^ = 0.05544, Sy. x = 0.5844), (**f**) The correlation between USS glucose with blood glucose (R^2^ = 0.8492, Sy. x = 0.2236). Sy. x means standard error of estimate (also seen as SEE).

**Figure 5 ijerph-19-04122-f005:**
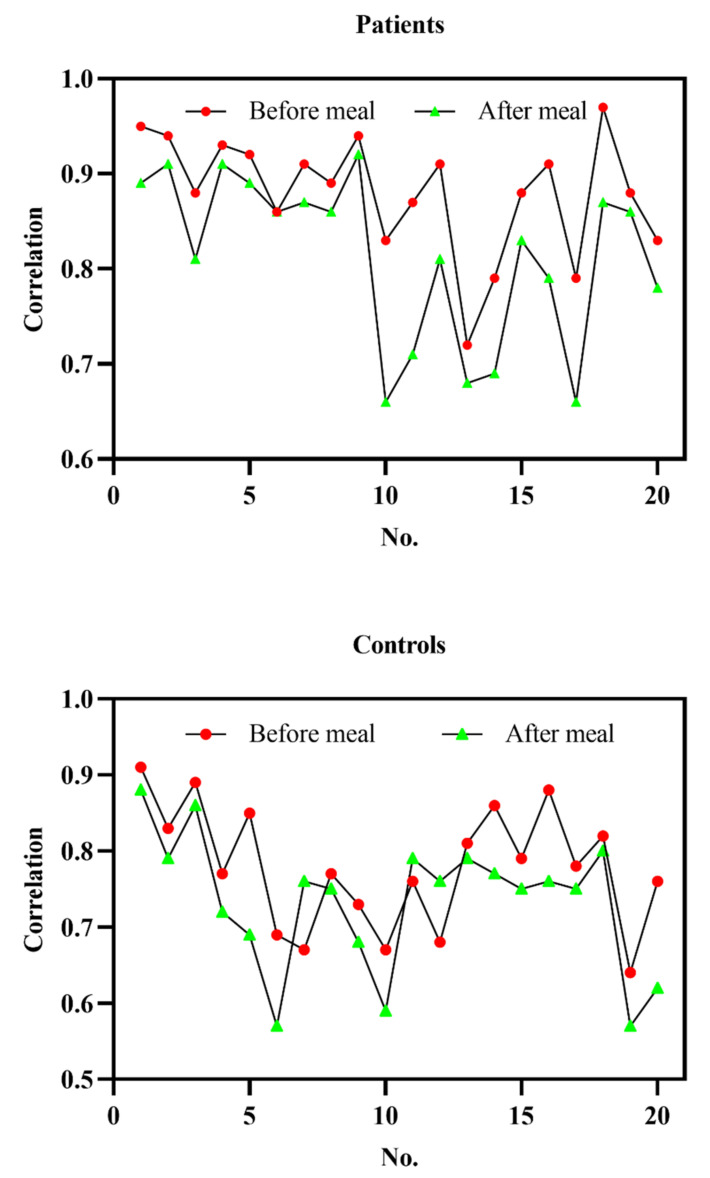
Correlation coefficients before and after meal of each group include DM patients and healthy controls, the Correlation refers to the correlation between blood and saliva glucose, and No. refers to the number of participants.

**Table 1 ijerph-19-04122-t001:** Saliva flow rate of the studied groups (DM patient/control and stimulated/unstimulated).

Salivary Flow (mL/min)	Patients (*n* = 40)		Controls (*n* = 40)	
	Male	Female	Male	Female
UWS	0.85 ± 0.18	0.85 ± 0.11	1.64 ± 0.16	0.89 ± 0.09
SWS	1.73 ± 0.12	1.52 ± 0.13	2.28 ± 0.15 **	1.78 ± 0.12
UPS	0.44 ± 0.09	0.43 ± 0.11 *	0.82 ± 0.08	0.45 ± 0.07
SPS	1.05 ± 0.09	0.91 ± 0.03	1.16 ± 0.08	0.91 ± 0.06
USS	0.72 ± 0.11	0.71 ± 0.09	1.41 ± 0.12	0.76 ± 0.08
SSS	1.47 ± 0.04	1.31 ± 0.33	1.94 ± 0.09	1.51 ± 0.14

** Maximum saliva flow. * Minimum saliva flow. UWS: unstimulated whole saliva, SWS: stimulated whole saliva, USS: unstimulated sublingual/submandibular saliva, SSS: stimulated sublingual/submandibular saliva, SPS: stimulated parotid saliva, UPS: unstimulated parotid saliva.

**Table 2 ijerph-19-04122-t002:** The average levels of saliva glucose for each group (DM patient/control and stimulated/unstimulated).

Glucose Levels (mM)	SWS	UWS	SPS	UPS	SSS	USS
Patients (*n* = 40)	1.53 ± 0.63	2.42 ± 0.66	1.84 ± 0.64	2.89 ± 0.76 **	1.79 ± 0.63	2.58 ± 0.52
Controls (*n* = 40)	1.42 ± 0.60 *	1.43 ± 0.40	1.50 ± 0.53	1.49 ± 0.49	1.54 ± 0.54	1.82 ± 0.31

** Maximum level of saliva glucose. * Minimum level saliva glucose. UWS: unstimulated whole saliva, SWS: stimulated whole saliva, USS: unstimulated sublingual/submandibular saliva, SSS: stimulated sublingual/submandibular saliva, SPS: stimulated parotid saliva, UPS: unstimulated parotid saliva.

**Table 3 ijerph-19-04122-t003:** Weekly correlation coefficient between DM patients and control group within one month.

No.	Patients (*n* = 10)					No.	Controls (*n* = 10)				
	1 w	2 w	3 w	4 w	CV %		1 w	2 w	3 w	4 w	CV %
1	0.93	0.89	0.94	0.89	2.8	1	0.91	0.89	0.87	0.83	3.9
2	0.92	0.87	0.89	0.91	2.4	2	0.82	0.81	0.79	0.76	3.3
3	0.86	0.87	0.85	0.82	2.5	3	0.88	0.87	0.83	0.81	3.8
4	0.91	0.86	0.92	0.87	3.3	4	0.76	0.75	0.77	0.72	2.8
5	0.88	0.91	0.89	0.93	2.4	5	0.84	0.83	0.79	0.78	3.6
6	0.84	0.75	0.83	0.81	4.9	6	0.68	0.67	0.65	0.68	2.1
7	0.89	0.84	0.88	0.85	2.7	7	0.66	0.65	0.68	0.61	4.5
8	0.87	0.87	0.86	0.83	2.2	8	0.76	0.75	0.69	0.77	4.8
9	0.92	0.84	0.91	0.88	4.0	9	0.72	0.71	0.65	0.67	4.8
10	0.81	0.83	0.86	0.78	4.1	10	0.66	0.65	0.59	0.64	4.8

## Data Availability

The study did not report any data.

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
