# Peer review of "Correlations of Salivary and Blood Glucose Levels among Six Saliva Collection Methods"

_ijerph, 2022, doi:10.3390/ijerph19074122_

Round 1

Reviewer 1 Report

It would be good to provide some background clinical information on the the diabetic patients with regards to the control of their diabetes and their medication.

Lines 57-58, This sentence needs revision as the the word acupuncture may convey a different meaning from blood testing.

Fig. !. Is there an error in the labelling of the Stimulated Parotid Saliva? It should be labelled SPS and not UPS (yellow box at end of line 154)

Line 178. There appears to be an error in the sentence as Table 1 does not show the blood and saliva glucose levels but the saliva flow rates in DM and control groups.

It will be helpful if the authors provides practical details of how saliva can be collected from the unstimulated parotid gland for routine testing.

Author Response

Ref: ijerph-1649579

Title: Correlation of glucose levels in saliva and blood among six saliva collection methods

Dear editor and reviewers:

Thank you very much for your letter and advice.

We appreciate editor and reviewers very much for their positive and constructive comments and suggestions on our manuscript entitled “Correlation of glucose levels in saliva and blood among six saliva collection methods” (Manuscript ID: ijerph-1649579). We have studied the comments of both the editor and reviewers carefully and tried our best to revise the paper accordingly. Point-to-point replies are included as below.

We would like to re-submit the revised manuscript for your consideration. I hope that the revision is acceptable for publication in your journal. Looking forward to hearing from you soon.

With best regards,

Yours sincerely,

Weiqiang Liu

************************************************************

List of replies

Dear editor and reviewers:

We would like to express our sincere thanks to you for the constructive and positive comments on our manuscript entitled “Correlation of glucose levels in saliva and blood among six saliva collection methods” (Manuscript ID: ijerph-1649579). The revisions have been made accordingly and highlighted in the revised manuscript with tracked changes.

Reviewers' comments:

Reviewer #1:

Comments and Suggestions for Authors

Point 1: It would be good to provide some background clinical information on the diabetic patients with regards to the control of their diabetes and their medication.

Response 1: Thank you for your careful review and valuable suggestions. The background clinical information on the diabetic patients with regards to the control of their diabetes and their medication were added, as followed. (Lines 52-58).

In this time frame, people may develop deadly problems over time, including retinal vascular disease, foot ulcers, renal failure, and other types of multiple organ damage [6]. However, the origin and pathogenesis of DM remain unknown, and there are no therapies available at the moment. The primary clinical control strategies include self-monitoring of blood glucose, medical nutrition therapy, exercise therapy, DM patient education, and DM patient medication therapy. Self-monitoring of blood glucose is critical for DM control and serves as the foundation for all strategies [7]. Sim et al. also [8] points out that frequent monitoring of blood glucose is of great significance for patients to grasp the condition and control the development of their complications. 

Point 2: Lines 57-58, This sentence needs revision as the word acupuncture may convey a different meaning from blood testing.

Response 2: Thank you for your careful review and valuable suggestions. This sentence is a bit redundant, repeating with the previous sentence. Now, it was substituted by adding the mental factor to the previous sentence, as followed. (Lines 62-64).

Unfortunately, the traditional blood glucose monitoring methods usually require blood sampling operations. Blood extraction is painful, inconvenient, brings the risk of infection and mental pains to patients, especially for those patients who need to check their blood glucose levels several times a day [9].

Point 3: Fig. !. Is there an error in the labelling of the Stimulated Parotid Saliva? It should be labelled SPS and not UPS (yellow box at end of line 154)

Response 3: Thank you for your careful review and valuable suggestions. There is an error in the labelling of the stimulated parotid saliva, it was corrected to SPS. As shown in Figure 1.

Figure 1. Six methods for collecting saliva samples, (a) UWS/SWS: the swab in the test tube was taken out and put in the mouth to chew for 3 min, (b) UPS/SPS: the swab was placed near the left parotid duct and it was taken out after 3 min, (c) USS /SSS: the swab was put under the tongue and it was taken out after 3 min.

Point 4: Line 178. There appears to be an error in the sentence as Table 1 does not show the blood and saliva glucose levels but the saliva flow rates in DM and control groups.

Response 4: Thank you for your careful review and valuable suggestions. There is an error in the sentence as Table 1 does not show the blood and saliva glucose levels but the saliva flow rates in DM and control groups. The sentence was corrected to ‘The blood and saliva flow rates of the studied groups are shown in Table 1(Line 186)’. Besides, we added the average of saliva glucose levels in different collection methods, as shown in Table 2. (Lines 199-200).

The blood and saliva flow rates of the studied groups are shown in Table 1. As can be observed, DM patient has a much lower saliva flow rate than the control group, and stimulated saliva has a significantly greater saliva flow rate than unstimulated saliva. Additionally, the patient group had greater salivary glucose levels (average levels) than the control group in all six saliva (as shown in Table 2), with the highest level of UPS glucose and the lowest level of stimulated entire salivary glucose, as illustrated in Figure 2. The patient group's mean blood glucose and salivary glucose levels were greater than the control group. Salivary glucose levels in the patient group ranged from 0.67 to 4.31 mmol/L. Salivary glucose levels in the control group ranged between 0.51 and 3.02 mmol/L. Salivary glucose levels increased in both the patient and control groups in correlation with blood glucose levels, as shown in Figure 4.

Table 2. The average levels of saliva glucose for each group (DM patient/control and stimulated / unstimulated).

Glucose levels (mM)

SWS

UWS

SPS

UPS

SSS

USS

Patients(n=40)

1.53±0.63

2.42±0.66

1.84±0.64

2.89±0.76**

1.79±0.63

2.58±0.52

Controls(n=40)

1.42±0.60*

1.43±0.40

1.50±0.53

1.49±0.49

1.54±0.54

1.82±0.31

** Maximum level of saliva glucose. * Minimum level saliva glucose.

Point 5: It will be helpful if the authors provide practical details of how saliva can be collected from the unstimulated parotid gland for routine testing.

Response 5: Thank you for your careful review and valuable suggestions. The practical details of how saliva can be collected from the stimulated and unstimulated parotid gland for routine testing was added in the caption of Figure 1.

Figure 1. Six methods for collecting saliva samples, (a) UWS/SWS: the swab in the test tube was taken out and put in the mouth to chew for 3 min, (b) UPS/SPS: the swab was placed near the left parotid duct and it was taken out after 3 min, (c) USS /SSS: the swab was put under the tongue and it was taken out after 3 min.

Reviewer 2 Report

SUMMARY/CONCEPT COMMENTS

The main objective of this work was to identify an ideal saliva collection method and use this method to determine the population and individual correlations between salivary glucose and blood glucose levels both in diabetes mellitus (DM) patients and controls. It also aimed to analyze the stability of the individual correlations. The importance of this work has been extensively explained in the manuscript, where the authors highlight the relevance of early detection of DM and the benefits of a non-invasive saliva-based monitoring method. Six diverse saliva collection methods were tested, and the outcomes were compared with the results from blood testing. The methods described are correct, although a bigger population would be preferable (as correctly stated by the authors at the end of the discussion section). The results obtained are promising, and the discussion is extensive and includes numerous relevant and generally recent references (with some legit auto-citations) that support the findings. The discussion would benefit from a restructuration and/or separation into smaller sections, and close parallelisms between this paper and a previous paper from the authors should be addressed for the sake of originality and novelty.

SPECIFIC COMMENTS

INTRODUCTION
Line 48-50

It would be good to clarify that it is an estimation, not an absolute truth. Maybe by saying “… around half of the 463 million adults with DM…”.

Line 50
Instead of a “however”, consider using an “and”, as the sentence does not contradict the previous, but complete it with further information.

Line 54
Written: “… All require..:”. Maybe better: “…usually require…”

Line 55
- Written: “…especially for blood glucose.” ? Redundant.
- Written: “…the method…”. Maybe better: “…blood extraction…”

Line 57
Written: “Besides the frequent…” I would eliminate that sentence (redundance). Maybe it can be substituted by adding the mental factor to the previous sentence in lines 55-56.

Line 59-60
Not just to improve the lives of diagnosed DM patients that need frequent monitoring, but also to ease preventive monitoring (early detection), as you mention previously.

Line 71
You say “Rodrigue”. It is “Rodrigues”.

Line 75
I would cancel the part “saliva sampling is challenging because” for the sake of clarity.

Line 87-90

Suggested change: “However, saliva cannot be considered a solitary fluid, as it consists in a complex mixture c the secretions of three major glands (the parotid, submandibular and sublingual glands, each secreting a characteristic type of saliva), hundreds of minor salivary glands, gingival crevicular fluid and debris.”

Line 90

It is written: “It is also not stable…”. Maybe better: “Moreover, saliva composition is constantly changing…”

Line 94

It is written: “Therefore…”. Maybe better: “Considering all this sampling-related composition variability, we can affirm that the saliva collection method and location will certainly have a great influence on the salivary glucose level”.

Paragraph lines 75-95

The order of the concepts is not very clear and can lead to misunderstandings. First, you mention the sampling issue. Afterwards you mention controversy in blood-saliva glucose correlation. Then you mention again the proved correlation among these sample types (already mentioned in previous paragraphs) and then you bring up the sampling issue again (including saliva sample heterogeneity) as a possible explanation of the blood-saliva lack of correlation observed in some works. I would restructure the paragraph in a way that the reader can follow the argument in an easier way.

Line 97

It is difficult to understand what the “respectively” refers to, as first 3 saliva types (whole saliva, parotid saliva and sublingual/mandibular saliva) are mentioned, and 2 methods (stimulated and unstimulated) are specified afterwards. If both methods were applied to the 3 saliva sampling types, maybe an alternative sentence would be: “In this study, whole saliva, parotid saliva and sublingual/mandibular saliva were collected with both stimulated and unstimulated methods, from 40…”

MATERIALS AND METHODS

Line 111

 “…participants: 40…”

Line 112
“…health and age..:”

Paragraph lines 119-126

- The paragraph would benefit from a reorganization, in order to explain in a clearer way to the reader the chronology of the methods and the aim: A first overall screening of the sampling possibilities, a further study using the most promising method(s)…

- When you say “for one week /for one month”, you mean everyday? Once per day? Weekly? More than that? Even if these are described further in the text and/or figures/tables, it is better to specify in the materials and methods section as well.

- The 10 DM patients and controls, where chosen from the remaining 20 individuals for each group or from the whole 40-individual group?

- What does “randomly” mean? It is important to distinguish randomly and arbitrarily. Was a computational/analogic method used to randomize such individuals?

Line 128
The paragraph begins abruptly. It could be begun with a sentence like: “The participants were asked to avoid smoking......... and not eating or drinking during the 30 minutes previous to sample collection. Before sampling, the mouth…”

It is worth it to clarify to the reader that “correlation” refers to “correlation between blood and saliva glucose”.

Line 144
Written: “mins”. The unit should be noted as “min”.

Line 148
Does “continuously” mean “daily”?

Line 149-150
Written: “before breakfast were collected”. Better: “… were collected before breakfast…”.

Line 164
- Please use the correct symbol for  by using the “equation” option in Word (code for LaTeX: \bar{x}).
- Standard deviation is usually noted as “SD”, not as “s”.

RESULTS

Line 171
“In the DM patient group…”

Line 174
Maybe better: “No statistically significant differences were observed due to gender… or age.”.

Line 178
I think that Table 1 does not show blood and salivary glucose levels, but salivary flow values.

Line 179-181
- “DM patients
- In order to make the statements more obvious, it would be worth to show the average values for each group (DM patient/control and stimulated/unstimulated).

Figure 2
- Please add A and B labels for the so-cited 2a and 2b figures, as well as more relevant information in the caption (e.g. error bar definition).
- The figure would be clearer if the legend was avoided and the X axis was labeled instead.
- Please consider using mM unit (in parenthesis) instead of mmol/L.
- Please consider using the “DM patients” label instead of just “patients” label.

Lines 181-184
It is written: “the salivary glucose level of the patient group was higher than control group in all six conditions, with the highest level of unstimulated parotid salivary glucose and the lowest stimulated whole salivary glucose level” is that true, considering the error bars? Have you assessed statistically significant differences? If so, please specify statistically-relevant data in the caption/text (e.g. p=0.05). Maybe a rearrangement of the figure would be useful to see that comparison (e.g. representing patient and control SWS results side by side, etc.).

Line 187
Written: ”Both in patient and control groups, salivary glucose levels increased with blood glucose levels.” Where is that shown? Please cite Figure 4, if applicable.

Lines 183-184
“…Fig. 2a and Fig. 2b…”

Line 192
“Fig. 3 shows” or “presents”.

Figure 4
Please define Sy.x

Line 228
“..:The overall stability was good” Please consider using a more scientifically precise word instead of “good”. “Acceptable”? “Inside the acceptable limits”?

Section 3.3.3.
The section would benefit from a grammar and syntax revision.

DISCUSSION + CONCLUSION

Line 243
“…one of the more widespread…”

Lines 249-250
One can either write “First” or “Firstly” (being “first” more accepted), but it is important to be consistent: After first, second. After firstly, secondly.

Paragraph lines 243-244
This is a summary of the introduction. Maybe this would not belong to the discussion section.

Line 262
“salivary glucose was found in both DM patients and control participants, which is consistent with the observations of Ephraim et al., who also found salivary glucose for both groups.”

Line 434
CV was already defined. “.. had a CV < 5%...”

GENERAL

- Tables and Figures would benefit from more descriptive captions. Refer the reader to the coloured on-line version when applicable (e.g. Fig. 3).
- A discussion section separated into smaller sections would be largely appreciated by the readers, and would help to highlight the main statements of the article.
- The paper would benefit from a close revision of typographic errors, grammar and syntax.

ETHICS/NOVELTY

There is parallelism between the present manuscript and the already-published work “Cui Y, Zhang H, Zhu J, et al. Unstimulated Parotid Saliva Is a Better Method for Blood Glucose Prediction[J]. Applied Sciences, 478 2021, 11(23): 11367“, from the authors of the present manuscript among others. For the sake of originality and novelty, I would suggest avoiding repetition of contents and results, including abstract, conclusions, descriptive figures, and captions. The paper would benefit from the transparent mention (not just citation) of the previous work (Ref [15] in the manuscript) and from the highlighting of the new findings in this manuscript in comparison with the previous one.

Author Response

Ref: ijerph-1649579

Title: Correlation of glucose levels in saliva and blood among six saliva collection methods

Dear editor and reviewers:

Thank you very much for your letter and advice.

We appreciate editor and reviewers very much for their positive and constructive comments and suggestions on our manuscript entitled “Correlation of glucose levels in saliva and blood among six saliva collection methods” (Manuscript ID: ijerph-1649579). We have studied the comments of both the editor and reviewers carefully and tried our best to revise the paper accordingly. Point-to-point replies are included as below.

We would like to re-submit the revised manuscript for your consideration. I hope that the revision is acceptable for publication in your journal. Looking forward to hearing from you soon.

With best regards,

Yours sincerely,

Weiqiang Liu

************************************************************

List of replies

Dear editor and reviewers:

We would like to express our sincere thanks to you for the constructive and positive comments on our manuscript entitled “Correlation of glucose levels in saliva and blood among six saliva collection methods” (Manuscript ID: ijerph-1649579). The revisions have been made accordingly and highlighted in the revised manuscript with tracked changes.

Reviewers' comments:

Reviewer #2:

SUMMARY/CONCEPT COMMENTS

Point 1: The main objective of this work was to identify an ideal saliva collection method and use this method to determine the population and individual correlations between salivary glucose and blood glucose levels both in diabetes mellitus (DM) patients and controls. It also aimed to analyze the stability of the individual correlations. The importance of this work has been extensively explained in the manuscript, where the authors highlight the relevance of early detection of DM and the benefits of a non-invasive saliva-based monitoring method. Six diverse saliva collection methods were tested, and the outcomes were compared with the results from blood testing. The methods described are correct, although a bigger population would be preferable (as correctly stated by the authors at the end of the discussion section). The results obtained are promising, and the discussion is extensive and includes numerous relevant and generally recent references (with some legit auto-citations) that support the findings. The discussion would benefit from a restructuration and/or separation into smaller sections, and close parallelisms between this paper and a previous paper from the authors should be addressed for the sake of originality and novelty.

Response 1: Thank you for your careful review and valuable suggestions. The discussion was restructured and/or separated into smaller sections, and close parallelisms between this paper and a previous paper from the authors were pointed out the sake of originality and novelty. (Lines 264-271 and Lines 276-280).

Saliva is considered an ultrafiltrate of blood and can replace blood for DM monitoring [17]. Among all salivary parameters, salivary glucose is most closely associated with the oral environment of DM patients [31]. However, the existence and degree of correlation between salivary glucose and blood glucose has been controversial. The main reason for this controversy is the influence of salivary glucose level. There are some factors as followed:

First, the saliva collection method will affect the salivary glucose level and the saliva flow rate, which has been confirmed in our previous study. This study pointed out that saliva collection method was an important factor that affected saliva glucose level and saliva flow rate. Besides, it pointed out UPS was the most correlated with blood glucose, which provided a reference for prediction of DM [17]. However, in our previous study, it is only for healthy people, and the saliva composition of DM patients is more complex, making it impossible to confirm whether the conclusions reached are also applicable to DM patients. 

Second, the population correlation can only reflect the general situation of the relationship between group blood glucose and salivary glucose. If it is used for the monitoring of the correlation between individual salivary glucose and blood glucose, its reliability has not been verified, and it is difficult to accurately reflect the blood glucose levels of different individuals at different stages. This has also not been thoroughly investigated in our previous study [17]. Therefore, in order to solve the above problems, this study compared the saliva flow rate and saliva glucose level of six different saliva collection methods in DM patients for the first time, explored the population correlation of saliva glucose and blood glucose. Besides, the saliva before and after breakfast were collected every day for one week to obtain the individual correlation. Finally, a month of follow-up monitoring was carried out to determine whether the individual correlation was stable, which laid a more scientific foundation for better monitoring the blood glucose level value with the salivary glucose level.

SPECIFIC COMMENTS

INTRODUCTION

Point 2: Line 48-50

It would be good to clarify that it is an estimation, not an absolute truth. Maybe by saying “… around half of the 463 million adults with DM…”.

Response 2: Thank you for your careful review and valuable suggestions. The sentence “… around half of the 463 million adults with DM…” was corrected to “Today, around half of the 463 million adults with DM are ignorant of their illness, placing them at an increased risk of acquiring catastrophic DM-related complications.” , as followed (Line 49).

Today, around half of the 463 million adults with DM are ignorant of their illness, placing them at an increased risk of acquiring catastrophic DM-related complications.

Point 3: Line 50

Instead of a “however”, consider using an “and”, as the sentence does not contradict the previous, but complete it with further information.

Response 3: Thank you for your careful review and valuable suggestions. “However” was changed with “Additionally” as followed. (Line 50).

Additionally, more recent research indicates that DM may present for up to seven years prior to clinical diagnosis [4,5]

Point 4: Line 54

Written: “… All require..:”. Maybe better: “…usually require…”

Response 4: Thank you for your careful review and valuable suggestions. “… All require..:” was changed with “…usually require…” as followed. (Line 61).

Unfortunately, the traditional blood glucose monitoring methods usually require blood sampling operations.

Point 5: Line 55

- Written: “…especially for blood glucose.” ? Redundant.

Response 5-1: Thank you for your careful review and valuable suggestions. The sentence of“…especially for blood glucose.” was Redundant. It has been deleted.

- Written: “…the method…”. Maybe better: “…blood extraction…”

Response 5-2: Thank you for your careful review and valuable suggestions. “…the method…” was changed with “…blood extraction…” as followed. (Line 62).

Blood extraction is painful, inconvenient, brings the risk of infection and mental pains to patients, especially for those patients who need to check their blood glucose levels several times a day [9].

Point 6: Line 57

Written: “Besides the frequent…” I would eliminate that sentence (redundance). Maybe it can be substituted by adding the mental factor to the previous sentence in lines 55-56.

Response 6: Thank you for your careful review and valuable suggestions. This sentence is a bit redundant, repeating with the previous sentence. Now, it was substituted by adding the mental factor to the previous sentence, as followed. (Lines 62-64).

Unfortunately, the traditional blood glucose monitoring methods usually require blood sampling operations. Blood extraction is painful, inconvenient, brings the risk of infection and mental pains to patients, especially for those patients who need to check their blood glucose levels several times a day [9].

Point 7: Line 59-60

Not just to improve the lives of diagnosed DM patients that need frequent monitoring, but also to ease preventive monitoring (early detection), as you mention previously.

Response 7: Thank you for your careful review and valuable suggestions. According to the comment, we added “but also to ease preventive monitoring (early detection)”, as followed. (Lines 65-67).

Not only a high need exists for a non-invasive glucose monitoring technology to make a major improvement in the lives of millions of people around the world living with DM, but also to ease preventive monitoring [10]

Point 8: Line 71

You say “Rodrigue”. It is “Rodrigues”.

Response 8: Thank you for your careful review and valuable suggestions. “Rodrigue” was changed with “Rodrigues” as followed. (Line 80).

According to Rodrigues et al. [21], saliva contains biomarkers such as different proteins, fatty acids, and carbs that, like blood, can reflect changes in human physiological activities, and so may serve as an alternative for early identification and monitoring of DM.

Point 9: Line 75

I would cancel the part “saliva sampling is challenging because” for the sake of clarity.

Response 9: Thank you for your careful review and valuable suggestions. For the sake of clarity, the sentence “saliva sampling is challenging because” was canceled according the comments.

Point 10: Line 87-90

Suggested change: “However, saliva cannot be considered a solitary fluid, as it consists in a complex mixture c the secretions of three major glands (the parotid, submandibular and sublingual glands, each secreting a characteristic type of saliva), hundreds of minor salivary glands, gingival crevicular fluid and debris.”

Response 10: Thank you for your careful review and valuable suggestions. The sentence was changed to “However, saliva cannot be considered a solitary fluid, as it consists in a complex mixture c the secretions of three major glands (the parotid, submandibular and sublingual glands, each secreting a characteristic type of saliva), hundreds of minor salivary glands, gingival crevicular fluid and debris.”, as followed. (Lines 92-96).

However, saliva cannot be considered a solitary fluid, as it consists in a complex mixture that comprises the secretions of three major glands (the parotid, submandibular and sublingual glands, each secreting a characteristic type of saliva), hundreds of minor salivary glands, gingival crevicular fluid and debris [27].

Point 11: Line 90

It is written: “It is also not stable…”. Maybe better: “Moreover, saliva composition is constantly changing…”

Response 11: Thank you for your careful review and valuable suggestions. The sentence was changed to “Moreover, saliva composition is constantly changing…” as followed. (Lines 96-97).

Moreover, saliva composition is constantly changing and the composition is affected among other things by saliva collection methods and general health.

Point 12: Line 94

It is written: “Therefore…”. Maybe better: “Considering all this sampling-related composition variability, we can affirm that the saliva collection method and location will certainly have a great influence on the salivary glucose level”.

Response 12: Thank you for your careful review and valuable suggestions. The sentence was changed to “Considering all this sampling-related composition variability, we can affirm that the saliva collection method and location will certainly have a great influence on the salivary glucose level” as followed. (Lines 99-101).

Considering all this sampling-related composition variability, we can affirm that the saliva collection method and location will certainly have a great influence on the salivary glucose level [28,29].

Point 13: Paragraph lines 75-95

The order of the concepts is not very clear and can lead to misunderstandings. First, you mention the sampling issue. Afterwards you mention controversy in blood-saliva glucose correlation. Then you mention again the proved correlation among these sample types (already mentioned in previous paragraphs) and then you bring up the sampling issue again (including saliva sample heterogeneity) as a possible explanation of the blood-saliva lack of correlation observed in some works. I would restructure the paragraph in a way that the reader can follow the argument in an easier way.

Response 13: Thank you for your careful review and valuable suggestions.

In order to let the reader can follow the argument in an easier way, we restructure the paragraph, as followed. (Lines 83-101).

Although the biomarkers in saliva reflect the health status of the human body, the use of salivary glucose as a diagnostic fluid for DM has been hindered, mainly because the correlation between salivary glucose and blood glucose has been greatly controversial [22]. Studies have shown that for DM patients, the salivary glucose level is positively correlated with the blood glucose level, so the salivary glucose can be used as a marker for DM detection [23,24]. However, in different studies, the relationship between salivary glucose and blood glucose is quite different with completely opposite conclusions, pointing out that salivary glucose and blood glucose have no significant correlation [25,26]. The key factor contributing to this phenomenon is that saliva is collected differently, and most studies view saliva wrongly as a homogeneous body fluid. However, saliva cannot be considered a solitary fluid, as it consists in a complex mixture that comprises the secretions of three major glands (the parotid, submandibular and sublingual glands, each secreting a characteristic type of saliva), hundreds of minor salivary glands, gingival crevicular fluid and debris [27]. Moreover, saliva composition is constantly changing and the composition is affected among other things by saliva collection methods and general health. The saliva mainly divided into whole saliva and single gland saliva, and saliva collection includes both stimulated and unstimulated methods. Considering all this sampling-related composition variability, we can affirm that the saliva collection method and location will certainly have a great influence on the salivary glucose level [28,29].

Point 14: Line 97

It is difficult to understand what the “respectively” refers to, as first 3 saliva types (whole saliva, parotid saliva and sublingual/mandibular saliva) are mentioned, and 2 methods (stimulated and unstimulated) are specified afterwards. If both methods were applied to the 3 saliva sampling types, maybe an alternative sentence would be: “In this study, whole saliva, parotid saliva and sublingual/mandibular saliva were collected with both stimulated and unstimulated methods, from 40…”

Response 14: Thank you for your careful review and valuable suggestions. Both methods were applied to the 3 saliva sampling types, so this part was changed to “In this study, whole saliva, parotid saliva and sublingual/mandibular saliva were collected with both stimulated and unstimulated methods, from 40…”, as followed. (Lines 102-104).

Therefore, in this study, whole saliva, parotid saliva and sublingual/mandibular saliva were collected with both stimulated and unstimulated methods, from 40 age-matched DM patients and 40 healthy controls with fasting state in the morning.

MATERIALS AND METHODS

Point 15: Line 111

 “…participants: 40…”

Response 15: Thank you for your careful review and valuable suggestions. The sentence was corrected, as followed. (Lines 119-120).

In this study, a total of 80 participants was included: 40 DM patients and 40 healthy controls.

Point 16: Line 112

“…health and age..:”

Response 16: Thank you for your careful review and valuable suggestions. The sentence was corrected, as followed. (Lines 120-121).

The inclusion criteria included: participants must be in good general health and≥18 years old.

Point 17: Paragraph lines 119-126

- The paragraph would benefit from a reorganization, in order to explain in a clearer way to the reader the chronology of the methods and the aim: A first overall screening of the sampling possibilities, a further study using the most promising method(s)…

Response 17-1: Thank you for your careful review and valuable suggestions. In order to explain in a clearer way to the reader the chronology of the methods and the aim, this paragraph has been restructured, as followed. (Lines 126-139).

Firstly, saliva and blood were collected from 80 participants, and the population correlation analysis was carried out to determine the ideal saliva collection method in the morning (7:30-8:00). Secondly, 20 DM patients and 20 healthy controls were arbitrarily selected from the 40 individuals for each group to collect saliva and blood every day (before and after breakfast) for one week with the determined saliva collection method, and conduct an individual correlation study. Finally, 10 DM patients and 10 controls were arbitrarily selected from the remaining 20 individuals for each group for continuous monitoring every day (before breakfast) for one month, and the weekly correlation of the individual was calculated, and the coefficient of variation (CV) was calculated to analyze the stability of the individual correlation. A first overall screening of the sampling possibilities is to choose the ideal saliva collection method, a further study using the most promising method to determine the correlation of individual relationships and their stability. The correlation in this study refers to the correlation between blood and saliva glucose.

- When you say “for one week /for one month”, you mean everyday? Once per day? Weekly? More than that? Even if these are described further in the text and/or figures/tables, it is better to specify in the materials and methods section as well.

Response 17-2: Thank you for your careful review and valuable suggestions. The sentence “for one week /for one month” means everyday, once per day. It was specified in the materials and methods section, as followed. (Line 129 and Line 133).

Firstly, saliva and blood were collected from 80 participants, and the population correlation analysis was carried out to determine the ideal saliva collection method in the morning (7:30-8:00). Secondly, 20 DM patients and 20 healthy controls were arbitrarily selected from the 40 individuals for each group to collect saliva and blood every day (before and after breakfast) for one week with the determined saliva collection method, and conduct an individual correlation study. Finally, 10 DM patients and 10 controls were arbitrarily selected from the remaining 20 individuals for each group for continuous monitoring every day (before breakfast) for one month, and the weekly correlation of the individual was calculated, and the coefficient of variation (CV) was calculated to analyze the stability of the individual correlation. A first overall screening of the sampling possibilities is to choose the ideal saliva collection method, a further study using the most promising method to determine the correlation of individual relationships and their stability. The correlation in this study refers to the correlation between blood and saliva glucose.

- The 10 DM patients and controls, where chosen from the remaining 20 individuals for each group or from the whole 40-individual group?

Response 17-3: Thank you for your careful review and valuable suggestions. The 10 DM patients and controls were chosen from the remaining 20 individuals for each group. It was corrected as followed. (Lines 128-129 and Lines 131-132).

Firstly, saliva and blood were collected from 80 participants, and the population correlation analysis was carried out to determine the ideal saliva collection method in the morning (7:30-8:00). Secondly, 20 DM patients and 20 healthy controls were arbitrarily selected from the 40 individuals for each group to collect saliva and blood every day (before and after breakfast) for one week with the determined saliva collection method, and conduct an individual correlation study. Finally, 10 DM patients and 10 controls were arbitrarily selected from the remaining 20 individuals for each group for continuous monitoring every day (before breakfast) for one month, and the weekly correlation of the individual was calculated, and the coefficient of variation (CV) was calculated to analyze the stability of the individual correlation. A first overall screening of the sampling possibilities is to choose the ideal saliva collection method, a further study using the most promising method to determine the correlation of individual relationships and their stability. The correlation in this study refers to the correlation between blood and saliva glucose.

- What does “randomly” mean? It is important to distinguish randomly and arbitrarily. Was a computational/analogic method used to randomize such individuals?

Response 17-4: Thank you for your careful review and valuable suggestions. In this study, we do not use computational/analogic method to randomize such individuals. We misused the randomly, it has been corrected to arbitrarily, as followed. (Line 128 and Line 132).

Firstly, saliva and blood were collected from 80 participants, and the population correlation analysis was carried out to determine the ideal saliva collection method in the morning (7:30-8:00). Secondly, 20 DM patients and 20 healthy controls were arbitrarily selected from the 40 individuals for each group to collect saliva and blood every day (before and after breakfast) for one week with the determined saliva collection method, and conduct an individual correlation study. Finally, 10 DM patients and 10 controls were arbitrarily selected from the remaining 20 individuals for each group for continuous monitoring every day (before breakfast) for one month, and the weekly correlation of the individual was calculated, and the coefficient of variation (CV) was calculated to analyze the stability of the individual correlation. A first overall screening of the sampling possibilities is to choose the ideal saliva collection method, a further study using the most promising method to determine the correlation of individual relationships and their stability. The correlation in this study refers to the correlation between blood and saliva glucose.

Point 18: Line 128

- The paragraph begins abruptly. It could be begun with a sentence like: “The participants were asked to avoid smoking......... and not eating or drinking during the 30 minutes previous to sample collection. Before sampling, the mouth…”

Response 18-1: Thank you for your careful review and valuable suggestions. This part has been restructured as followed. (Lines 141-143).

The participants were asked to avoid smoking, brushing teeth and not eating or drinking during the 30 minutes previous to sample collection. Before sampling, the mouth was rinsed with water before collection to remove food residues in the oral cavity [30].

- It is worth it to clarify to the reader that “correlation” refers to “correlation between blood and saliva glucose”.

Response 18-2: Thank you for your careful review and valuable suggestions. The meaning of “correlation” refers to “correlation between blood and saliva glucose” was pointed out as followed. (Lines 138-139).

Firstly, saliva and blood were collected from 80 participants, and the population correlation analysis was carried out to determine the ideal saliva collection method in the morning (7:30-8:00). Secondly, 20 DM patients and 20 healthy controls were arbitrarily selected from the 40 individuals for each group to collect saliva and blood every day (before and after breakfast) for one week with the determined saliva collection method, and conduct an individual correlation study. Finally, 10 DM patients and 10 controls were arbitrarily selected from the remaining 20 individuals for each group for continuous monitoring every day (before breakfast) for one month, and the weekly correlation of the individual was calculated, and the coefficient of variation (CV) was calculated to analyze the stability of the individual correlation. A first overall screening of the sampling possibilities is to choose the ideal saliva collection method, a further study using the most promising method to determine the correlation of individual relationships and their stability. The correlation in this study refers to the correlation between blood and saliva glucose.

Point 19: Line 144

Written: “mins”. The unit should be noted as “min”.

Response 19: Thank you for your careful review and valuable suggestions. The “mins” was corrected to“min” as followed. (Line 157 and Line 158).

Venous blood was drawn from all subjects following the last saliva sample. The samples were gently mixed for 1 min and then placed immediately on ice for 30 min. After centrifuging the samples at 1000 g for 15 min at 4°C, the upper two-thirds aliquot of plasma was frozen at -70°C until analysis.

Point 20: Line 148

Does “continuously” mean “daily”?

Response 20: Thank you for your careful review and valuable suggestions. Yes, in this part “continuously” means “daily”, it was changed as followed. (Line 161 and Line 163).

Individual correlation study: UPS and blood were collected before and after breakfast, other processing conditions and storage methods were unchanged, and collected daily for one week. Individual correlation stability study: UPS and blood were collected before breakfast, other processing conditions and storage methods remained unchanged, and were collected daily for one month. Glucose and saliva glucose were measured by GOD-POD (Glucose kit, Beijing Furui Runkang Biotechnology Co., Ltd., China).

Point 21: Line 149-150

Written: “before breakfast were collected”. Better: “… were collected before breakfast…”.

Response 21: Thank you for your careful review and valuable suggestions. The sentence was changed to “… were collected before breakfast…”, as followed. (Line 162).

Individual correlation study: UPS and blood were collected before and after breakfast, other processing conditions and storage methods were unchanged, and collected daily for one week. Individual correlation stability study: UPS and blood were collected before breakfast, other processing conditions and storage methods remained unchanged, and were collected daily for one month. Glucose and saliva glucose were measured by GOD-POD (Glucose kit, Beijing Furui Runkang Biotechnology Co., Ltd., China).

Point 22: Line 164

- Please use the correct symbol for by using the “equation” option in Word (code for LaTeX: \bar{x}).

- Standard deviation is usually noted as “SD”, not as “s”.

Response 22: Thank you for your careful review and valuable suggestions. The correct symbol was used for by using the “equation” option in Word. Besides, Standard deviation was corrected to “SD”, as followed. (Line 173 and Line 175).

To do statistical analysis, Graphpad 8.0 was employed. The data from the enumeration were expressed as relative numbers, and the  test was used to compare groups. The measurement data were normally distributed and reported as (mean standard deviation), with the t test used to compare groups. Salivary glucose levels of different genders and ages were determined in each group using the t-test on two independent samples. The receiver operating characteristic (ROC) approach was utilized in this study to completely evaluate the diagnostic utility of salivary glucose detection in DM. P < 0.05 indicates a statistically significant difference.

RESULTS

Point 23: Line 171

“In the DM patient group…”

Response 23: Thank you for your careful review and valuable suggestions. The sentence was corrected to “There were 14 males and 26 females with an average age of (50.1 ± 4.8) years in DM patient group”. (Line 182).

There were 14 males and 26 females with an average age of (50.1 ± 4.8) years in DM patient group and 14 males and 26 females with an average age of (49.7 ± 3.7) years in control group. No statistically significant differences were observed due to gender (t = 0.641, P = 0.289) or age (t = 0.181, P = 0.510) between the two groups.

Point 24: Line 174

Maybe better: “No statistically significant differences were observed due to gender… or age.”.

Response 24: Thank you for your careful review and valuable suggestions. The sentence was changed to “No statistically significant differences were observed due to gender (t = 0.641, P = 0.289) or age (t = 0.181, P = 0.510) between the two groups.”, as followed. (Lines 184-185).

There were 14 males and 26 females with an average age of (50.1 ± 4.8) years in DM patient group and 14 males and 26 females with an average age of (49.7 ± 3.7) years in control group. No statistically significant differences were observed due to gender (t = 0.641, P = 0.289) or age (t = 0.181, P = 0.510) between the two groups.

Point 25: Line 178

I think that Table 1 does not show blood and salivary glucose levels, but salivary flow values.

Response 25: Thank you for your careful review and valuable suggestions. There is an error in the sentence as Table 1 does not show the blood and saliva glucose levels but the saliva flow rates in DM and control groups. The sentence was corrected to ‘The blood and saliva flow rates of the studied groups are shown in Table 1(Line 186)’. Besides, we added the average of saliva glucose levels in different collection methods, as shown in Table 2. (Line 199).

The blood and saliva flow rates of the studied groups are shown in Table 1. As can be observed, DM patient has a much lower saliva flow rate than the control group, and stimulated saliva has a significantly greater saliva flow rate than unstimulated saliva. Additionally, the patient group had greater salivary glucose levels (average levels) than the control group in all six saliva (as shown in Table 2), with the highest level of UPS glucose and the lowest level of stimulated entire salivary glucose, as illustrated in Figure 2. The patient group's mean blood glucose and salivary glucose levels were greater than the control group. Salivary glucose levels in the patient group ranged from 0.67 to 4.31 mmol/L. Salivary glucose levels in the control group ranged between 0.51 and 3.02 mmol/L. Salivary glucose levels increased in both the patient and control groups in correlation with blood glucose levels, as shown in Figure 4.

Table 2. The average levels of saliva glucose for each group (DM patient/control and stimulated / unstimulated).

Glucose levels (mM)

SWS

UWS

SPS

UPS

SSS

USS

Patients(n=40)

1.53±0.63

2.42±0.66

1.84±0.64

2.89±0.76**

1.79±0.63

2.58±0.52

Controls(n=40)

1.42±0.60*

1.43±0.40

1.50±0.53

1.49±0.49

1.54±0.54

1.82±0.31

** Maximum level of saliva glucose. * Minimum level saliva glucose.

Point 26: Line 179-181

- “DM patients”

- In order to make the statements more obvious, it would be worth to show the average values for each group (DM patient/control and stimulated/unstimulated).

Response 26-1: Thank you for your careful review and valuable suggestions. The average values for each group (DM patient/control and stimulated/unstimulated) were added in Table 2, as followed.

Table 2. The average levels of saliva glucose for each group (DM patient/control and stimulated / unstimulated).

Glucose levels (mM)

SWS

UWS

SPS

UPS

SSS

USS

Patients(n=40)

1.53±0.63

2.42±0.66

1.84±0.64

2.89±0.76**

1.79±0.63

2.58±0.52

Controls(n=40)

1.42±0.60*

1.43±0.40

1.50±0.53

1.49±0.49

1.54±0.54

1.82±0.31

** Maximum level of saliva glucose. * Minimum level saliva glucose.

Figure 2

- Please add A and B labels for the so-cited 2a and 2b figures, as well as more relevant information in the caption (e.g. error bar definition).

- The figure would be clearer if the legend was avoided and the X axis was labeled instead.

- Please consider using mM unit (in parenthesis) instead of mmol/L.

- Please consider using the “DM patients” label instead of just “patients” label.

Response 26-2: Thank you for your careful review and valuable suggestions. Figure 2 was corrected as shown in Figure 2, as followed:

(1)A and B labels for the so-cited 2a and 2b figures were added, as well as more relevant information in the caption; (2) Legend was avoided and the X axis was labeled instead of legend; (3) The mM unit (in parenthesis) was used instead of mmol/L; (4) The “DM patients” was used label instead of just “patients” label. (5) The “Healthy controls” was used label instead of just “controls” label.

Figure 2. Salivary glucose levels in all participants, a) DM patient group, B) Healthy control group.

Point 27: Lines 181-184

It is written: “the salivary glucose level of the patient group was higher than control group in all six conditions, with the highest level of unstimulated parotid salivary glucose and the lowest stimulated whole salivary glucose level” is that true, considering the error bars? Have you assessed statistically significant differences? If so, please specify statistically-relevant data in the caption/text (e.g. p=0.05). Maybe a rearrangement of the figure would be useful to see that comparison (e.g. representing patient and control SWS results side by side, etc.).

Response 27: Thank you for your careful review and valuable suggestions. In order to solve the problems, we added the average of saliva glucose levels in six different collection methods, as shown in Table 2.

The blood and saliva flow rates of the studied groups are shown in Table 1. As can be observed, DM patient has a much lower saliva flow rate than the control group, and stimulated saliva has a significantly greater saliva flow rate than unstimulated saliva. Additionally, the patient group had greater salivary glucose levels (average levels) than the control group in all six saliva (as shown in Table 2), with the highest level of UPS glucose and the lowest level of stimulated entire salivary glucose, as illustrated in Figure 2. The patient group's mean blood glucose and salivary glucose levels were greater than the control group. Salivary glucose levels in the patient group ranged from 0.67 to 4.31 mmol/L. Salivary glucose levels in the control group ranged between 0.51 and 3.02 mmol/L. Salivary glucose levels increased in both the patient and control groups in correlation with blood glucose levels, as shown in Figure 4.

Table 2. The average levels of saliva glucose for each group (DM patient/control and stimulated / unstimulated).

Glucose levels (mM)

SWS

UWS

SPS

UPS

SSS

USS

Patients(n=40)

1.53±0.63

2.42±0.66

1.84±0.64

2.89±0.76**

1.79±0.63

2.58±0.52

Controls(n=40)

1.42±0.60*

1.43±0.40

1.50±0.53

1.49±0.49

1.54±0.54

1.82±0.31

** Maximum level of saliva glucose. * Minimum level saliva glucose.

Point 28: Line 187

Written: ”Both in patient and control groups, salivary glucose levels increased with blood glucose levels.” Where is that shown? Please cite Figure 4, if applicable.

Response 28: Thank you for your careful review and valuable suggestions. “Both in patient and control groups, salivary glucose levels increased with blood glucose levels.” was shown in Figure 4, it was cited now, as followed. (Lines 197-198).

The blood and saliva flow rates of the studied groups are shown in Table 1. As can be observed, DM patient has a much lower saliva flow rate than the control group, and stimulated saliva has a significantly greater saliva flow rate than unstimulated saliva. Additionally, the patient group had greater salivary glucose levels (average levels) than the control group in all six saliva (as shown in Table 2), with the highest level of UPS glucose and the lowest level of stimulated entire salivary glucose, as illustrated in Figure 2. The patient group's mean blood glucose and salivary glucose levels were greater than the control group. Salivary glucose levels in the patient group ranged from 0.67 to 4.31 mmol/L. Salivary glucose levels in the control group ranged between 0.51 and 3.02 mmol/L. Salivary glucose levels increased in both the patient and control groups in correlation with blood glucose levels, as shown in Figure 4.

Point 29: Lines 183-184

“…Fig. 2a and Fig. 2b…”

Response 29: Thank you for your careful review and valuable suggestions. Figure 2 was corrected as shown in Figure 2, as followed:

(1)A and B labels for the so-cited 2a and 2b figures were added, as well as more relevant information in the caption; (2) Legend was avoided and the X axis was labeled instead of legend; (3) The mM unit (in parenthesis) was used instead of mmol/L; (4) The “DM patients” was used label instead of just “patients” label. (5) The “Healthy controls” was used label instead of just “controls” label.

Figure 2. Salivary glucose levels in all participants, a) DM patient group, B) Healthy control group.

Point 30: Line 192

“Fig. 3 shows” or “presents”.

Response 30-1: Thank you for your careful review and valuable suggestions. It was changed to” Figure 3 showed… ”, as followed. (Line 207).

Figure 3 showed the ROC curve of salivary glucose in the diagnosis of DM. It can be seen that the salivary glucose obtained by unstimulated method can significantly distinguish the DM patients from the control group, and the P values were all < 0.001.

Figure 4

Please define Sy.x

Response 30-2: Thank you for your careful review and valuable suggestions. The definition of Sy.x was added as followed. (Line 228).

Methods

SWS

UWS

SPS

UPS

SSS

USS

R2

0.001127

0.8109

0.1568

0.9153*

0.05544

0.8492

Sy.x

0.6183

0.3217

0.5597

0.2772

0.5844

0.2236

* Maximum correlation; Sy. x means standard error of estimate (also seen as SEE).

Figure 4. Correlation between salivary glucose of six saliva collection method and blood glucose.

Point 31: Line 228

“..:The overall stability was good” Please consider using a more scientifically precise word instead of “good”. “Acceptable”? “Inside the acceptable limits”?

Response 31-1: Thank you for your careful review and valuable suggestions. The sentence ..:The overall stability was good” was changed a more scientifically precise word instead of “good”, as followed. (Lines 239-241).

It can be seen from the population correlation that the UPS had the highest glucose level and was most correlated with the blood glucose level. Besides, it can be better used to diagnose DM. Therefore, the UPS was used to study the individual correlation between blood glucose and salivary glucose. The average correlation coefficient between pre-prandial salivary glucose and blood glucose in DM patients was 0.88, and the average correlation coefficient between postprandial salivary glucose and blood glucose was 0.813, while the average correlation coefficient between pre-prandial salivary glucose and blood glucose in the control group was 0.78. The average correlation coefficient between postprandial salivary glucose and blood glucose was 0.7325. As shown in Figure 5, while the correlation coefficients before and after breakfast for various individuals within a week varied significantly, the overall consistency was high. In general, the correlation coefficients before breakfast were higher than those after breakfast.

Section 3.3.3.

The section would benefit from a grammar and syntax revision.

Response 31-2: Thank you for your careful review and valuable suggestions. The section has been revised from a grammar and syntax, as followed. (Lines 248-255).

In section 3.3.2, we can conclude that correlation before breakfast was stronger and more stable in the individual correlation study. So, the saliva before breakfast was used to examine the stability of the correlation coefficient. The average daily pre-breakfast salivary glucose and blood glucose levels were tracked consistently for one month under the assumption that all other experimental circumstances and techniques were constant. Finally, for each subject, the correlation coefficient between pre-prandial salivary glucose and blood glucose levels within a week was computed. Table 3 summarized the specific findings. As can be observed, all results had a CV < 5%.

DISCUSSION + CONCLUSION

Point 32: Line 243

“…one of the more widespread…”

Response 32: Thank you for your careful review and valuable suggestions. This is a summary of the introduction. It does not belong to the discussion section. It has been deleted.

Point 33: Lines 249-250

One can either write “First” or “Firstly” (being “first” more accepted), but it is important to be consistent: After first, second. After firstly, secondly.

Response 33-1: Thank you for your careful review and valuable suggestions. This part was changed to be consistent: First, second, as followed. (Line 264 and Line 272).

First, the saliva collection method will affect the salivary glucose level and the saliva flow rate, which has been confirmed in our previous study. This study pointed out that saliva collection method was an important factor that affected saliva glucose level and saliva flow rate. Besides, it pointed out UPS was the most correlated with blood glucose, which provided a reference for prediction of DM [17]. However, in our previous study, it is only for healthy people, and the saliva composition of DM patients is more complex, making it impossible to confirm whether the conclusions reached are also applicable to DM patients.

Second, the population correlation can only reflect the general situation of the relationship between group blood glucose and salivary glucose. If it is used for the monitoring of the correlation between individual salivary glucose and blood glucose, its reliability has not been verified, and it is difficult to accurately reflect the blood glucose levels of different individuals at different stages. This has also not been thoroughly investigated in our previous study [17]. Therefore, in order to solve the above problems, this study compared the saliva flow rate and saliva glucose level of six different saliva collection methods in DM patients for the first time, explored the population correlation of saliva glucose and blood glucose. Besides, the saliva before and after breakfast were collected every day for one week to obtain the individual correlation. Finally, a month of follow-up monitoring was carried out to determine whether the individual correlation was stable, which laid a more scientific foundation for better monitoring the blood glucose level value with the salivary glucose level.

Paragraph lines 243-244

This is a summary of the introduction. Maybe this would not belong to the discussion section.

Response 33-2: Thank you for your careful review and valuable suggestions. Yes, this is a summary of the introduction. It does not belong to the discussion section. It has been deleted.

Point 34: Line 262

“salivary glucose was found in both DM patients and control participants, which is consistent with the observations of Ephraim et al., who also found salivary glucose for both groups.”

Response 34: Thank you for your careful review and valuable suggestions. The sentence was changed to “salivary glucose was found in both DM patients and control participants, which is consistent with the observations of Ephraim et al., who also found salivary glucose for both groups.”, as followed. (Lines 285-287)

In this study, salivary glucose was found in both DM patients and healthy controls, which is consistent with the observations of Ephraim et al. [32] who also found salivary glucose for both groups.

Point 35: Line 434

CV was already defined. “.. had a CV < 5%...”

Response 35: Thank you for your careful review and valuable suggestions. CV was all corrected to“.. had a CV < 5%...” throughout the manuscript, e.g. as followed. (Lines 454-455).

In this study, the salivary flow rate and salivary glucose levels of six different salivary collection methods were compared, and the correlations between individual salivary glucose and blood glucose were explored. A month-long follow-up monitoring yielded the stability of individual correlations. For all six saliva collection methods, the mean salivary glucose levels in DM patients were greater than those in the control group. When comparing stimulated and unstimulated saliva, stimulated saliva glucose levels decreased and saliva flow increased. It was found that UPS glucose level was most correlated with blood glucose level. The AUC was 0.9316, which could accurately distinguish DM patients. The correlation coefficient between saliva glucose and blood glucose in different DM patients was quite different. The average pre-prandial correlation coefficient was 0.83, and the postprandial correlation coefficient was 0.77. Besides, the pre-prandial correlation coefficient had a CV < 5% within 1 month. In conclusion, based on the findings of this study, it can be inferred that UPS before breakfast may serve as a potential non-invasive adjunct to monitoring blood glucose in DM patients.

GENERAL

Point 36: - Tables and Figures would benefit from more descriptive captions. Refer the reader to the coloured on-line version when applicable (e.g. Fig. 3).

Response 36: Thank you for your careful review and valuable suggestions. Tables and Figures were added more descriptive captions, as followed.

Figure 1. Six methods for collecting saliva samples, (a) UWS/SWS: the swab in the test tube was taken out and put in the mouth to chew for 3 min, (b) UPS/SPS: the swab was placed near the left parotid duct and it was taken out after 3 min, (c) USS /SSS: the swab was put under the tongue and it was taken out after 3 min. (Lines 167-170).

Figure 2. Salivary glucose levels in all participants, a) DM patient group, B) Healthy control group. (Lines 204-205).

Figure 3. The ROC curve of salivary glucose in the diagnosis of DM of six saliva collection methods. (Lines 219-220).

Figure 4. Correlation between salivary glucose of six saliva collection method and blood glucose. (Line 229).

Figure 5. Correlation coefficients before and after meal of each group include DM patients and healthy controls, the Correlation refers to the correlation between blood and saliva glucose, and NO. refers to the number of participants. (Lines 244-246).

Table 1. Saliva flow rate of the studied groups (DM patient/control and stimulated/unstimulated). (Line 186).

Table 2. The average levels of saliva glucose for each group (DM patient/control and stimulated/unstimulated). (Lines 199-200).

Table 3. Weekly correlation coefficient between DM patients and control group within one month. (Line 256).

Point 37: - A discussion section separated into smaller sections would be largely appreciated by the readers, and would help to highlight the main statements of the article.

Response 37: Thank you for your careful review and valuable suggestions. In order to highlight the main statements of the article. The discussion section was separated into smaller sections. E. g. as followed. (Lines 258-296).

Saliva is considered an ultrafiltrate of blood and can replace blood for DM monitoring [17]. Among all salivary parameters, salivary glucose is most closely associated with the oral environment of DM patients [31]. However, the existence and degree of correlation between salivary glucose and blood glucose has been controversial. The main reason for this controversy is the influence of salivary glucose level. There are some factors as followed:

First, the saliva collection method will affect the salivary glucose level and the saliva flow rate, which has been confirmed in our previous study. This study pointed out that saliva collection method was an important factor that affected saliva glucose level and saliva flow rate. Besides, it pointed out UPS was the most correlated with blood glucose, which provided a reference for prediction of DM [17]. However, in our previous study, it is only for healthy people, and the saliva composition of DM patients is more complex, making it impossible to confirm whether the conclusions reached are also applicable to DM patients.

Second, the population correlation can only reflect the general situation of the relationship between group blood glucose and salivary glucose. If it is used for the monitoring of the correlation between individual salivary glucose and blood glucose, its re-liability has not been verified, and it is difficult to accurately reflect the blood glucose levels of different individuals at different stages. This has also not been thoroughly investigated in our previous study [17]. Therefore, in order to solve the above problems, this study compared the saliva flow rate and saliva glucose level of six different saliva collection methods in DM patients for the first time, explored the population correlation of saliva glucose and blood glucose. Besides, the saliva before and after breakfast were collected every day for one week to obtain the individual correlation. Finally, a month of follow-up monitoring was carried out to determine whether the individual correlation was stable, which laid a more scientific foundation for better monitoring the blood glucose level value with the salivary glucose level.

In this study, salivary glucose was found in both DM patients and healthy controls, which is consistent with the observations of Ephraim et al. [32] who also found salivary glucose for both groups. However, few like Amer et al. [33] found no salivary glucose of the control group. We found that the saliva glucose levels of DM patients obtained by the six saliva collection methods were higher than control group, and the difference was statistically significant, which was consistent with the conclusions of most foreign studies [23,24]. Similar to our study, Mishra et al. [34] mentioned a positive and statistically significant correlation between salivary and blood glucose in DM patients. Therefore, salivary glucose can be used to predict blood glucose level in DM patients. Karjalainen et al. [35] have shown that after good blood glucose control in DM patients, both salivary glucose and blood glucose are reduced to varying degrees, which suggests that salivary glucose and blood glucose have a certain correlation.

Point 38: - The paper would benefit from a close revision of typographic errors, grammar and syntax.

Response 38: Thank you for your careful review and valuable suggestions. The typographic errors, grammar and syntax, etc. had been revised in revised manuscript.

ETHICS/NOVELTY

Point 39: There is parallelism between the present manuscript and the already-published work “Cui Y, Zhang H, Zhu J, et al. Unstimulated Parotid Saliva Is a Better Method for Blood Glucose Prediction[J]. Applied Sciences, 478 2021, 11(23): 11367“, from the authors of the present manuscript among others. For the sake of originality and novelty, I would suggest avoiding repetition of contents and results, including abstract, conclusions, descriptive figures, and captions. The paper would benefit from the transparent mention (not just citation) of the previous work (Ref [15] in the manuscript) and from the highlighting of the new findings in this manuscript in comparison with the previous one.

Response 39: Thank you for your careful review and valuable suggestions.

The discussion was restructured and/or separated into smaller sections, and close parallelisms between this paper and a previous paper from the authors were pointed out the sake of originality and novelty. (Lines 264-271 and Lines 276-280).

Saliva is considered an ultrafiltrate of blood and can replace blood for DM monitoring [17]. Among all salivary parameters, salivary glucose is most closely associated with the oral environment of DM patients [31]. However, the existence and degree of correlation between salivary glucose and blood glucose has been controversial. The main reason for this controversy is the influence of salivary glucose level. There are some factors as followed:

First, the saliva collection method will affect the salivary glucose level and the saliva flow rate, which has been confirmed in our previous study. This study pointed out that saliva collection method was an important factor that affected saliva glucose level and saliva flow rate. Besides, it pointed out UPS was the most correlated with blood glucose, which provided a reference for prediction of DM [17]. However, in our previous study, it is only for healthy people, and the saliva composition of DM patients is more complex, making it impossible to confirm whether the conclusions reached are also applicable to DM patients. 

Second, the population correlation can only reflect the general situation of the relationship between group blood glucose and salivary glucose. If it is used for the monitoring of the correlation between individual salivary glucose and blood glucose, its reliability has not been verified, and it is difficult to accurately reflect the blood glucose levels of different individuals at different stages. This has also not been thoroughly investigated in our previous study [17]. Therefore, in order to solve the above problems, this study compared the saliva flow rate and saliva glucose level of six different saliva collection methods in DM patients for the first time, explored the population correlation of saliva glucose and blood glucose. Besides, the saliva before and after breakfast were collected every day for one week to obtain the individual correlation. Finally, a month of follow-up monitoring was carried out to determine whether the individual correlation was stable, which laid a more scientific foundation for better monitoring the blood glucose level value with the salivary glucose level.

Round 2

Reviewer 2 Report

Dear Authors,

You did a great and fast job with the modifications of the paper.
I have no further observations.

This manuscript is a resubmission of an earlier submission. The following is a list of the peer review reports and author responses from that submission.